# Hemocyanins: Microscopic Giants with Unique Structural Features for Applications in Biomedicine

**DOI:** 10.3390/vaccines13111086

**Published:** 2025-10-23

**Authors:** Michelle L. Salazar, Diego A. Díaz-Dinamarca, Javier Bustamante, Felipe Vergara, Augusto Manubens, Fabián Salazar, María Inés Becker

**Affiliations:** 1Fundación Ciencia y Tecnología para el Desarrollo (FUCITED), Santiago 7750269, Chile; 2Pathology Advanced Translational Research Unit (PATRU), Department of Pathology and Laboratory Medicine, Emory University School of Medicine, Atlanta, GA 30322, USA; 3Medical Research Council Centre for Medical Mycology, University of Exeter, Exeter EX4 4QD, UK; 4Departamento de Investigación y Desarrollo, Biosonda Corp., Santiago 7750629, Chile

**Keywords:** mollusk hemocyanins, immunomodulators, protein-based adjuvants, Th1 immune response, innate immune receptors, antigenic processing

## Abstract

Vaccine adjuvants play a crucial role in the field of vaccinology, yet they remain one of the least developed and poorly characterized components of modern biomedical research. The limited availability of clinically approved adjuvants highlights the urgent need for new molecules with well-defined mechanisms and improved safety profiles. Hemocyanins, large copper-containing metalloglycoproteins found in mollusks, represent a unique class of natural immunomodulators. Hemocyanins serve as carrier proteins that help generate antibodies against peptides and hapten molecules. They also function as non-specific protein-based adjuvants (PBAs) in both experimental human and veterinary vaccines. Their mannose-rich N-glycans allow for multivalent binding to innate immune receptors, including C-type lectin receptors (e.g., MR, DC-SIGN) and Toll-like receptor 4 (TLR4), thereby activating both MyD88- and TRIF-dependent signaling pathways. Hemocyanins consistently favor Th1-skewed immune responses, which is a key characteristic of their adjuvant potential. Remarkably, their conformational stability supports slow intracellular degradation and facilitates dual routing through MHC-II and MHC-I pathways, thereby enhancing both CD4+ and CD8+ T-cell responses. Several hemocyanins are currently being utilized in biomedical research, including Keyhole limpet hemocyanin (KLH) from *Megathura crenulata*, along with those from other gastropods such as *Concholepas concholepas* (CCH), *Fissurella latimarginata* (FLH), *Rapana venosa* (RvH), and *Helix pomatia* (HpH), all of which display strong immunomodulatory properties, making them promising candidates as adjuvants for next-generation vaccines against infectious diseases and therapeutic immunotherapies for cancer. However, their structural complexity has posed challenges for their recombinant production, thus limiting their availability from natural sources. This reliance introduces variability, scalability issues, and challenges related to regulatory compliance. Future research should focus on defining the hemocyanin immunopeptidome and isolating minimal peptides that retain their adjuvant activity. Harnessing advances in structural biology, immunology, and machine learning will be critical in transforming hemocyanins into safe, reproducible, and versatile immunomodulators. This review highlights recent progress in understanding how hemocyanins modulate mammalian immunity through their unique structural features and highlights their potential implications as potent PBAs for vaccine development and other biomedical applications. By addressing the urgent need for novel immunostimulatory platforms, hemocyanins could significantly advance vaccine design and immunotherapy approaches.

## 1. Introduction

Interest in the use of mollusk hemocyanins for biomedical purposes was sparked in the 1960s, when the remarkable immunogenic and immunostimulatory properties of those from *Megathura crenulata*—also known as keyhole limpet hemocyanins (KLHs)—were introduced. These hemocyanins were used in clinical applications to assess human immunocompetence through a KLH delayed-type hypersensitivity (DTH) test [1,2]. Later animal model studies confirmed their immunomodulatory properties [3,4]. With its colossal structure and xenogenic origin, KLH quickly became the preferred carrier protein for developing poly- and monoclonal antibodies against peptides and hapten molecules in experimental animals [5,6]; today, it is used as a model antigen for human toxicological and immune challenge studies [7,8,9,10]. These applications were later expanded to utilize KLH as a non-specific immunomodulatory agent that is clinically approved in Argentina, Austria, the Netherlands, and South Korea for preventing superficial bladder carcinoma recurrence after tumor resection [11,12,13], as an immunogenic neoantigen that stimulates patients’ effective Th1 immune response [14,15], KLH has been approved for developing therapeutic agents. It has an excellent safety record and is available for GMP-grade products, which are evaluated in early-phase trials as investigational medicinal products. As a carrier protein, KLH enhances antigen immunogenicity by combining adjuvant properties with the ability to trigger innate immunity through localized proinflammatory responses at the injection site [16]. These features make KLH a preferred carrier choice for experimental cancer vaccines, where hemocyanin act as adjuvants to stimulate T-cell responses against weak antigens like tumor-associated antigens, such as the gangliosides GM2, GD2, and GloboH [17,18,19], as well as in drug abuse vaccine design, including for haptens as cocaine, nicotine, and opioids [18,20,21,22,23].

KLH has garnered notable attention for its use in antitumor vaccines, which offer an alternative to conventional antiproliferative therapies, such as chemotherapy and radiotherapy, either alone or in combination with these treatments. One specific application has involved the use of KLH in autologous anti-idiotypic vaccines for B-cell lymphomas. In this approach, the variable regions of the immunoglobulin expressed on the tumorigenic B-cell clone (Id) are chemically linked to KLH. Phase I/II clinical trials demonstrated that the Id-KLH vaccine combined with GM-CSF was effective in eliciting tumor-specific immune responses and achieving molecular remission in patients with stage 1 lymphoma [24,25,26]. More recently, an immunotherapy approach combining an antimyeloma Id-KLH vaccine with vaccine-primed costimulated T cells has been reported, yielding a robust immune response with Id-specific humoral and cellular responses compared to the control with KLH alone [27]. Furthermore, KLH also serves as an adjuvant that helps disrupt self-tolerance to tumor antigens. This immunotherapy involves generating ex vivo autologous tumor cell lysate-loaded dendritic cells and KLH that stimulate T-cell responses in cancer patients, showing promising outcomes in extending overall survival [28,29]. Beyond its anticancer applications, antimicrobial activity against the Gram-positive *Streptococcus epidermis* and the Gram-negative *Escherichia coli* has also been described in FUs of hemocyanins from *Helix* and *Rapana* snails [30].

The widespread applications of KLH, including its use in vaccines as a carrier and protein-based adjuvant (PBA) [16], have prompted the exploration of other hemocyanins that could complement or even replace it. This is partly because its supply depends on natural sources, and, to date, recombinant KLH expression has not been possible. However, this has not affected their primary applications. In fact, hemocyanins have advantages over other natural or therapeutically used antigens that are commonly used in experimental human immune challenges, such as varicella zoster or BCG. These traditional antigens do not allow for control over the degree, duration, and timing of primary exposure. The long history of using hemocyanins has established an excellent safety record, making them a safe and reliable model for immunomodulatory proteins. However, the limitations in the use of KLH are largely due to the methodological inconsistencies in their application, leading to a lack of standardized approaches in the design and conduct of early-phase drug development [15].

Nowadays, a panel of gastropod hemocyanins has been studied, and their well-established immunomodulatory properties have been preclinically evaluated both in vivo and in vitro. Among the most notable is CCH from *Concholepas concholepas* [31], which was declared the “International Mollusk of the Year 2023” by the LOEWE Center for Translational Biodiversity Genomics and the International Society for Mollusk Research. CCH has been evaluated as a carrier protein for producing antibodies [32,33,34] as an antitumor agent in animal models of superficial bladder carcinoma [35], melanoma [36,37,38], and oral squamous cell carcinoma [39]. It has also been used as a PBA in a Phase I clinical trial, and as an adjuvant in a tumor antigen-presenting cell vaccine for patients with castration-resistant prostate cancer [40,41]. Additionally, it is currently being evaluated in patients with advanced malignant melanoma. In addition, the use of CCH as a carrier/adjuvant has been well evaluated in immunocontraceptive vaccines for female deer, cats, bison, and horses [42,43,44,45,46]. Moreover, the Environmental Protection Agency (US EPA) approves formulations including KLH or CCH [47].

Other emergent gastropod hemocyanins that have been biochemically and immunologically characterized are FLH from *Fissurella latimarginata*, which demonstrated increased immunogenicity and antitumor activity in a melanoma model compared to KLH and CCH [48]; HpH from *Helix pomatia*, which has shown suitable potential as an adjuvant for subunit influenza and tetanus vaccines [49], as well as significant anticancer effects in murine models of colon carcinoma and melanoma [50,51]; RvH from *Rapana venosa*, which has exhibited potential as an adjuvant and immunotherapeutic agent against hematological malignancies [52,53]; and, most recently, PcH from *Pomacea canaliculata*, which induces a proinflammatory effect on a human macrophage cell lines and promotes the phenotypic changes associated with monocyte to macrophage differentiation [54]. The Appendix A lists these known hemocyanins with their structural characteristics and examples of applications.

Nowadays, hemocyanins are understood as models of exogenous thymus-dependent antigens, i.e., able to induce antibody production with the help of T CD4+ lymphocytes. Moreover, it has been established that the non-specific immunotherapeutic effects of KLH and CCH in superficial bladder cancer rely on adequate priming with these hemocyanins [35,36,55]. This fact highlights the significance of the adaptive immune response. Accordingly, several authors have attributed the favorable immunomodulatory properties of hemocyanins in mammals to their xenogeneic origin, large size, and complex, oligosaccharide-stabilized structure [6,51,56,57]. Indeed, hemocyanins are giant molecules that exhibit some of the most complex known quaternary structures [58].

## 2. Hemocyanin Exhibits Distinctive Structural Characteristics

Understanding the molecular basis and design of hemocyanins is crucial for future biomedical and biotechnological applications. Early studies focused on dissociating hemocyanin molecules by increasing the pH, adding denaturing agents, or removing divalent cations. This dissociation was followed by reassociation, which yielded valuable insights into their structural features. Initial findings were supported by negative staining observed through transmission electron microscopy and later confirmed by X-ray crystallography and sequence analysis. These studies provided comprehensive information about the shape, organization, and notable structural properties of hemocyanins [58,59,60].

Hemocyanins from gastropods have a molecular mass of approximately 8 MDa or higher, making them some of the largest proteins in the animal kingdom. These glycoproteins are easily visible under a transmission electron microscope with negative staining, which reveals large, hollow, cylindrical molecules, as shown in Figure 1. The quaternary structure of these hemocyanins is schematized in Figure 2, which depicts the decamer—the basic structure of which has a diameter of approximately 310 Å and a height of approximately 160 Å—formed by ten subunits that are not covalently linked, with masses ranging from 350 to 450 kDa which can be associated as didecamers and, in some cases, as tri- and even multidecamers [59,61,62]. These subunits can combine along the rotational axis to form a cylinder with a diameter of approximately 35 nm and a height of 18 nm, which is not hollow but partially filled with a structure designated as the collar [59]. At the same time, the subunits consist of a string of eight paralogous oxygen-binding globular folded domains called functional units (FUs), ranging from 45 to 50 kDa each. These FUs, designed as FU-a to FU-h, are arranged sequentially from the N-terminal FU and linked on a single polypeptide chain with a peptide linker of approximately 15 amino acid residues [63,64]. FU-a to FU-f contribute to the cylinder wall, whereas FU-g and FU-h form the internal collar complex [64,65]. As a result, the structural organization of mollusk hemocyanins provides them with high conformational stability, which has been experimentally demonstrated in studies examining their stability against pH variations, thermal changes, and chemical denaturation [54,66,67,68]. Limited proteolysis and the identification of FUs using immunological techniques have enabled the precise FU ordering across different species. Each FU possesses a binuclear active site with two copper ions complexed by six histidine residues, an arrangement classified as a type 3 copper center. As their Hill coefficients indicate, hemocyanins bind oxygen with low cooperativity [62,69]. When this type 3 center binds an O_2_ molecule, it forms a Cu_2_O_2_ cluster, generating light absorbance around 340 nm and thus oxygenated hemocyanin’s characteristic blue color (a coloration that deoxygenated hemocyanin does not exhibit).

In gastropods, didecamers can be formed with a single (homodidecamer) or with two types of subunits (heterodidecamers), with D5 point group symmetry [59,70]. In KLH, HpH, HtH, and RvH hemocyanins, which consist of two types of subunits, the molecules are organized as homodecamers. Consequently, as homodidecamers, each molecule contains single subunits which coexist in the animal’s circulation [5,67,71,72,73]. In KLH, the subunits are different and do not share common epitopes. In contrast, CCH subunits contain common and specific epitopes, forming heterodecamers and, consequently, heterodidecamers, i.e., molecules composed of two types of intermingled subunits [31,74]. FLH has a single subunit type, forming homodecamers and -didecamers [48].

Mollusk hemocyanins have another significant structural feature: they contain predominantly diverse N-glycans (Asn-linked), which exhibit unique monosaccharide compositions and variable side-chain structures that are not found in normal mammalian cells [75]. These glycosylations include, for instance, a new type of N-glycan with Gal(β1-6)Man-motifs [76,77], and the novel Gal(β1-4)Gal(β1-4)Fuc(alpha1-6)-core modification attached to the proximal N-acetylglucosamine, both in KLH [78]. Moreover, the unusual, truncated N-glycosylation pattern, characterized by only two mannoses and an internal fucose, is found in CCH [79], as well as the novel acidic terminal glycan on RvH [75,80]. The N-glycans in hemocyanins can comprise up to 9% (*w*/*w*) of the molecule and contribute to the quaternary structure’s formation and stabilization. Indeed, N-deglycosylation affects the association among subunits and thus impairs this quaternary structure [68,81]. O-glycosylations have also been reported (Ser/Thr/Tyr-linked) in KLH and CCH [82,83].

Remarkably, hemocyanins’ glycosylation trees are similar to those found on the surface of tumor cells and pathogens. For instance, the Thomsen–Friedenreich antigen disaccharide core 1 O-glycan, Galβ1-3GalNAcα1-O-Ser/Thr (T antigen), is found in KLH [84] and CCH [68]. It has thus been proposed that in superficial bladder cancer, which expresses this oligosaccharide, antibodies against hemocyanin can trigger a cross-reaction and promote the immune response’s effector mechanism against tumor cells and pathogens [84,85], such as the activation of the classical pathway of the human complement system [86]. Additionally, structures bearing an α1,6-linked fucose and a β1,2 xylose linked on the core pentasaccharide—characteristics of parasites such as *Schistosoma mansoni* [78,85], as well as the motif Gal(β1-6)-Man with oligosaccharide side chains from bacteria like *Salmonella typhimurium* and *Klebsiella pneumoniae*—have also been reported in hemocyanins [76].

Hemocyanin sequences from several gastropod species are now available, and crystal structures for different FUs are known [63,65,79,87]. Although attempts have been made to recombinantly express the FUs of both subunits of KLH in *Escherichia coli*, their structure and immunogenicity have not been thoroughly characterized [88]. Consequently, a heterologous full-length hemocyanin has not yet been successfully produced, likely due to its high molecular mass and complex quaternary structure. Furthermore, the proper linkage of specific carbohydrate side chains at the appropriate N- and O-glycosylation sites is crucial for correct hemocyanin folding, which will be vital for its future clinical applications.

## 3. Hemocyanin Size and Structure Disassembly Has No Significant Impact on Immunogenicity, but N-Deglycosylation Does

Our preliminary studies aimed to determine whether the size, quaternary structure, or specific hemocyanin sequences are essential for activating the mammalian immune system. To explore these properties, we analyzed the immune response to individual CCH subunits (CCH-A and CCH-B). Each of these accounted for approximately one-twentieth of the entire molecule after disassembling the quaternary structure. Surprisingly, the isolated subunits were immunogenic and exhibited varying antitumor effects in a murine melanoma model. The CCH-A subunit was particularly immunogenic and demonstrated a more potent antitumor effect. These findings led us to compare the two subunits, revealing that CCH-A contained a higher oligosaccharide content [83]. In turn, this prompted us to study the role of glycosylation in hemocyanins’ immunomodulatory effects. Additionally, studies of *Rapana thomasiana* hemocyanin indicated that the whole molecule and its isolated subunits (RtH1 and RtH2) were strongly immunogenic in mice [52]. In contrast, research on dissociated KLH subunits has shown that they are immunogenic but less effective than the whole KLH in immunotherapy experiments [71]. These results suggest that whole hemocyanin molecules or their subunits have variable immunomodulatory properties, provided they retain glycosylation.

We have demonstrated that N-deglycosylated forms of CCH, FLH, and KLH disrupt their quaternary structure, significantly decreasing their immunogenic effects in both in vitro and in vivo studies. This effect on their structure reduces the production of proinflammatory cytokines, such as TNF, IL-6, and IL-12p40, in in vitro-cultured macrophages. In a murine model study, lower levels of antibodies were found [68]. Furthermore, deglycosylated FLH fails to induce the Th1 robust murine DC maturation characteristic of hemocyanins, resulting in decreased levels of IL-6 and IL-12p40 compared to the control [48]. Additionally, a significant decrease in TNF secretion by murine macrophages was observed with deglycosylated PcH [54]. These results suggest that glycosylation plays a crucial role in hemocyanin-induced antigen-presenting cell (APC) maturation. To further support these findings, we used trypsin to partially digest the FLH, creating smaller fragments while keeping glycosylation intact. The results indicated that the smaller FLH fragments maintained their non-specific immunostimulatory effects, as well as that the DCs’ maturation was dose dependent [48]. Moreover, the hemocyanin-mediated proinflammatory cytokine response was reduced in monocyte-derived human DCs when treated with deglycosylated FLH and KLH [89]. These observations are supported by the fact that hemocyanin glycosylations interact with various innate immune receptors localized on APCs, such as mouse and human MR (mannose receptor, CD 206) and human DC-SIGN (DC-specific ICAM-3-grabbing nonintegrin, CD 209) in a carbohydrate-dependent manner [68,89]. The interaction between FLH and KLH with MGL (macrophage galactose-type lectin, CD 301) has been examined using ELISA with a chimeric receptor (MGL-Fc). While the results indicate that these hemocyanins interact with MGL [68], this finding has not been validated in APCs, unlike other innate immunity receptors that have been studied. It is important to note that CCH, KLH, and FLH’s interactions with these receptors and their proinflammatory immune response differ, which can be partly attributed to oligosaccharide content and composition variation among these hemocyanins. Indeed, their carbohydrate contents are 3.1%, 3.4%, and 4.1% (*w*/*w*), respectively. Interestingly, compared to CCH and KLH, FLH, which has a higher carbohydrate content, is more immunogenic and exhibits increased antitumor activity in a murine melanoma model [48].

We now know that when native hemocyanins are inoculated, they induce a robust Th1-type immune response against them, exerting a bystander activator effect in neighboring cells, breaking tumor tolerance and favoring latent immune reactions occurring in an individual, augmenting IFNg-production, and enhancing in vivo antitumor activity [90,91] (Figure 3). Indeed, hemocyanins induce a significant proinflammatory cytokine milieu that drives gene expression towards M1 macrophages secreting IL-6, TNF-α, and IFN-γ with immunostimulatory Th1-orienting properties [48,92], activating NK cells against tumor cells [39,93]. It is important to note that early immunohistochemical studies of biopsies from patients with transitional cell carcinoma treated with KLH reveal significant cellular activation nine months after therapy initiation. This activation is characterized by an intense mononuclear cell and CD4+ lymphocyte infiltration, along with smaller numbers of CD8+ T cells and granulocytes [94]. These findings suggest that the effects of KLH may be closely linked to non-specific immune system stimulation, which could lead to the development of an antitumor response. In this context, we cannot dismiss the possibility that these T CD8+ cells are bystander T cells, which are known—through indirect mechanisms such as cytokine signaling or innate receptor activation—to become activated during an ongoing immune response without directly recognizing their specific antigen [95]. However, whether these mechanisms play a role in hemocyanin-induced immunomodulatory effects has not yet been investigated.

## 4. Hemocyanins’ Endocytosis Through Binding to Innate Immune Receptors Provides Insights into How They Activate the Immune System

Our main goal has been to understand how hemocyanins activate the immune system. These glycoproteins bind to innate immune endocytic receptors, providing valuable insights into immune responses. Transmission electron microscopy analyses have shown that mouse and human DCs primarily use micropinocytosis for hemocyanin uptake. Additionally, receptor-mediated endocytosis also contributes to this process, as electron microscopy images reveal that CCH, KLH, and FLH molecules are associated with coated pits and vesicles in mouse and human APCs [36,89]. Based on these observations, we hypothesized that APCs would recognize hemocyanins as a “highly mannosylated molecular pattern” through C-type lectin receptor (CLR) family, which are involved in recognizing endogenous and carbohydrate-associated pathogens in a calcium-dependent manner [96]. This hypothesis was supported by the discovery that an antibody blocking the MR partially inhibited KLH-mediated activation of human DCs [97]. It is important to note that CLRs expressed by APCs play a significant role in initiating and regulating various immune processes. The high densities of terminal mannose and GlcNAc residues on hemocyanins can thus stimulate their phagocytosis through CLRs on APCs. Once inside the cells, hemocyanins are delivered to different cellular compartments, where they bind to class I and II major histocompatibility complex (MHC-I and MHC-II) molecules, activating CD8+ and CD4+ T lymphocytes, respectively. The receptor-mediated endocytosis of hemocyanins would therefore modulate antigen presentation, effectively initiating the adaptive immune response, which is a well-studied process [98,99,100].

In this context, we initially focused on two significant mannose-recognizing innate immune CLRs: MR [101] and DC-SIGN [102]. The results showed that human and mouse APCs partially internalize hemocyanins in a calcium-dependent manner, leading to proinflammatory cytokine secretion. Moreover, we demonstrated for the first time that these clinically significant hemocyanins—KLH and CCH—directly bind to the carbohydrate recognition domains of these receptors with affinity constants within the physiological concentration range and colocalize with these receptors after being internalized into human DCs through clathrin-mediated endocytosis [89].

Unexpectedly, we also discovered that Toll-like receptor 4 (TLR4, CD284) is involved in the hemocyanin-mediated proinflammatory response [103]. In fact, in vitro analysis showed that all three native hemocyanins (CCH, FLH, and KLH) bind to the TLR4 chimeric receptor in a dose-dependent manner. This relationship significantly diminishes when they are deglycosylated. The involvement of TLR4 was confirmed with a TLR4-blocking antibody in APCs, in DCs derived from the murine strain C3H/HeJ (which is nonfunctional for this receptor), and in TLR4-deficient macrophages. Hemocyanin-induced proinflammatory cytokine secretion was reduced in all these cultures compared to their respective controls. Furthermore, we demonstrate that KLH and FLH induce ERK1/2 kinase phosphorylation, which is a key event in the TLR4 signaling pathway [103]. While the primary function of TLR4 is not endocytosis, it can collaborate with other receptors to facilitate the translocation of peptides from endosomes to the cytosol, as observed with MR and DC-SIGN [98,104]. Indeed, TLR4 works with MR, lacking the cytoplasmic domain necessary to transduce the external signal into intracellular pathways [105,106]. Moreover, MR can directly influence the activation of various immune cells, playing a critical role in regulating this process and shaping inflammatory responses and antigen presentation [107,108]. MR specifically targets antigens in early endosomes rather than lysosomes, allowing for their presentation via MHC-I through a process known as cross-presentation, which is essential for responding to virus-infected and tumor cells [109]. In contrast, DC-SIGN agonists are directed to lysosomes for MHC-I and MHC-II processing [110], indicating their involvement in hemocyanin signaling pathways. However, CCH, KLH, and FLH do not interact with other CLRs that bind to *mannose*, such as Dectin-1 and Dectin-2 [103].

## 5. As Antigens of Higher Conformational Stability, Hemocyanins Are Slowly Processed into MHC-II and, Surprisingly, MHC-I Pathways to Drive Th1 Immune Responses

Research on the antigen processing of small proteins, such as ovalbumin (OVA), has been extensively conducted. However, similar studies on larger and structurally stable oligomeric metalloglycoproteins, such as hemocyanins, are still limited [111]. As mentioned earlier, the glycosylation of hemocyanins contributes to their immunological effects by interacting with receptors involved in innate immunity. Interestingly, even deglycosylated hemocyanins and isolated subunits exhibit partial immunostimulant effects, indicating that additional structural features, such as quaternary structure or sequence, may contribute to their beneficial effects. Hidden linear or conformational immunodominant epitopes may exist within the hemocyanin molecule, as described for CCH using monoclonal antibodies [74]. However, the existence of other nondominant epitopes—named cryptic or subdominant epitopes—that become exposed during intracellular protein processing in APCs can elicit alternative immune responses [112,113]. Indeed, the position of an epitope within the immunodominance hierarchy can be influenced by various factors related to determinant selection in APCs. These factors include antigen processing, TAP-dependent peptide transport, and factors related to the responding T-cell population, such as the affinity of the T-cell receptor’s (TCR) interaction with the peptide-MHC complex, among others [114]. It is striking that despite the extensive use of KLH in biomedicine and biotechnology, there is a significant lack of research on its associated intracellular processing pathways. Similarly, data on CCH are limited. Thus, while glycosylations play a role in recognizing hemocyanins by innate immune receptors, the primary structure and intrinsic conformational stability of the epitopes presented may also contribute to their immunogenicity, particularly in vaccine development [115]. Additionally, glycosylation can influence T-cell recognition, allowing hemocyanin to produce immunogenic glycopeptides recognized by T-lymphocytes in an MHC-restricted manner. Research has shown that N-linked GlcNAc residues, which are abundant in hemocyanins, remain on the peptides that bind to MHC-II molecules after glycoprotein processing. These glycan groups can contribute to peptide structure conformation, in turn contributing to recognized T epitope formation [116,117,118]. Despite this, glycopeptides in hemocyanins have not been identified or experimentally evaluated; however, we have recently begun understanding their antigenic processing after APCs endocytose them.

Antigen-presenting cells process hemocyanins as exogenous antigens through the MHC-II pathway; however, this intracellular degradation occurs significantly more slowly than with other model protein antigens, such as ovalbumin (OVA). While OVA processing in vitro takes about one day, processing hemocyanin takes several [119]. This difference is significant, as APCs process proteins with higher conformational stability more slowly, which is often associated with enhanced immune responses [120]. This intrinsic protein property significantly impacts the kinetics of their proteolytic degradation and determines their intracellular fate, influencing immunogenicity and immune polarization. Research has demonstrated that stable proteins can resist acidification during lysosomal maturation, allowing them to escape into the cytoplasm of APCs and enter the cross-presentation pathway, where immunostimulatory environments shape the ensuing T-cell response [121,122]. This process is essential for vaccines that generate CD8+ cytotoxic T lymphocytes [123,124]. In this context, we unexpectedly discovered that hemocyanins are also processed through the MHC-I cross-presentation pathway [119,125]. Moreover, we have demonstrated that hemocyanins enhance CD8+ T-cell priming in OT-I mice when used as peptide-based OVA adjuvants (PBAs) [125]. One possible explanation for these results is that hemocyanin is an agonist of DC-SIGN, a receptor with a high affinity for multivalent glycans. This leads to its internalization and subsequent targeting into the lysosomal pathway, ultimately presenting specific peptides in MHC-II. Additionally, DC-SIGN can prime CD8+ T cells through an endocytic pathway, leading to cross-presentation [114,126]. Supporting this, proteasome inhibitors disrupt FLH’s adjuvant activity when co-administered with OVA, effectively abolishing its cross-presentation [125].

Our early studies on the cellular localization of hemocyanins in bone marrow-derived dendritic cells (BMDCs) revealed that hemocyanins are found in early endosomal compartments, identified by the small GTPase Rab5, as well as in late endosomal and lysosomal compartments (Rab7 and LAMP-1) [127]. Further analysis involving different pharmacological inhibitors showed that MG132 and epoxomicin—both proteasome inhibitors—significantly inhibited the secretion of IL-12p40. In contrast, bafilomycin, a lysosomal acidification inhibitor, significantly decreased IL-6 and IL-12p40 levels. Additionally, BMDCs treated with leupeptin and pepstatin—both cathepsin inhibitors—demonstrated that leupeptin inhibited IL-12p40 secretion. In contrast, pepstatin A inhibited both IL-12p40 and IL-6 secretion. These results indicate that the proteasomal and lysosomal pathways for cross-presentation are involved in hemocyanin-induced cytokine secretion [119,128].

In addition, observations regarding the kinetics of Th1 proinflammatory cytokines induced by hemocyanins, in association with lysosomes and proteasomes, indicate that they are stored in subcellular compartments for several days. Following this period, hemocyanin fragments, stimulated by the proinflammatory environment they create, are transferred to APC cytoplasm, where the proteasome can process them. This is supported by confocal microscopy, which has shown that hemocyanins are associated with these organelles for up to 96 h. In contrast, the degradation of OVA and its proinflammatory effects occur within 24 h. Furthermore, KLH and CCH colocalize with the lysosomal marker LAMP-1 for several days, contributing to the slow degradation process [119]. Inside these compartments, hemocyanins undergo slow proteolysis, which ensures a continuous supply of peptide-loaded MHC-I molecules over an extended period. This phenomenon has also been observed with other complex antigens stored in organelles similar to lysosome compartments [129,130].

## 6. Hemocyanins’ Interaction with TLR4 Promotes the TLR4 Signaling Pathways Associated with TRIF and MyD88 Adaptor Proteins

Investigating inflammatory pathways is crucial for comprehending the full spectrum of hemocyanin’s immunomodulatory effects in mammals, including its potential as an adjuvant and non-specific anticancer agent. By exploring these mechanisms, we can gain insight into how hemocyanins influence immune responses, activating both innate and adaptive immunity, by interacting with innate receptors on antigen-presenting cells, inducing expression of cosignalling molecules, triggering cytokine production, and influencing T helper cell activation, expansion and polarization. It is known that *CLRs* and TLRs activate distinct inflammatory pathways. Initial studies suggested that the signaling pathway by which hemocyanins promote the secretion of proinflammatory cytokines depends on the involvement of spleen tyrosine kinase (Syk kinase; CLRs couple Syk kinase activation) and ERK1/2, a key event in the TLR4 signaling pathways. Indeed, it has been shown that the pharmacological inhibition of Syk kinase with piceatannol reduces the secretion of proinflammatory cytokines, such as TNF, IL-6, and IL-12p40, in BMDCs cultured with FLH compared to controls without the inhibitor [103]. It was subsequently reported that KLH activated the transcription factor NF-κB in the human monocyte reporter cell line THP-1, an effect that was inhibited by the Syk kinase inhibitor BAY 61-3606. Moreover, KLH induces ERK1/2 kinase phosphorylation, and its pharmacological inhibition decreased NF-κB activation [131]. These results suggest that Syk kinase and ERK1/2 are associated with KLH-induced NF-κB activation in human monocytes. As mentioned previously, studies have shown that FLH and KLH induce the TLR4-dependent phosphorylation of ERK1/2 in murine DCs and macrophages, which aligns with the reported effect of KLH in THP-1 cells. However, in this study, Syk kinase phosphorylation was not detected. Moreover, DCs from mice deficient in the myeloid differentiation marker MyD88, a downstream adapter molecule of TLR4, which activates signaling molecules that turn on the NF-κB protein, were partially activated by FLH, strongly suggesting that the TLR pathway plays a role in hemocyanin recognition to activate APCs [103] (Figure 4).

Adjuvants that rely on TLR4 ligands—derived from nontoxic lipid A region of lipopolysaccharide (LPS), such as the Monophosphoryl lipid A (*MPLA*)—are used in human vaccines. This is because TLR4 has a unique ability among the TLR family to signal through the recruitment of two adapter proteins: MyD88 and Toll-Interleukin-1 Receptor (TIR) domain-containing Interferon-Beta Adapter (TRIF) [132,133]. Through MyD88, TLR4 activates transcription factors NF-κB and AP-1, which are associated with the rapid production of proinflammatory cytokines and the prompt initiation of innate immune responses to address infectious threats. In contrast, TRIF signaling pathways that result in heightened type I interferon expression are associated with the adaptive immune responses essential for effective vaccination [134,135]. Characterizing non-glycolipid TLR4 ligands, such as hemocyanins, can provide valuable mechanistic insights that may lead to the development of new adjuvant formulations.

In this context, we hypothesized that the recruitment of MyD88 and TRIF is essential for generating hemocyanins’ TLR4-dependent effects. We aimed to investigate whether the interaction between hemocyanins and TLR4 can induce differences in signaling pathways, and found distinct outcomes. Consequently, the potency of CCH, FLH, and KLH was analyzed in a dose–response study using HEK-blue reporter cells expressing either mouse (mTLR4) or human TLR4 (hTLR4). FLH showed a partial and weak agonist effect, respectively, while KLH and CCH both behaved as weak agonists. When we examined the impact of MyD88 on the immune response triggered by FLH in BMDCs using a MyD88 inhibitory peptide, we observed a decrease in the transcript levels of IL-6, CD80, CD86, and IFIT1, but not in COX-2 transcript levels. Similarly, when the impact of TRIF on the immune response triggered by FLH was examined using an inhibitory peptide, it reduced the transcription of IL-6, CD80, IFIT1, and IP-10 but not CD86. These data suggest that MyD88 and TRIF are crucial in the signaling patterns when TLR4 interacts with hemocyanins [125]. It is important to note that activating these signaling pathways is crucial for the successful adjuvant activity of TLR4 agonists, as well as for driving a robust Th1 adaptive immune response [136].

## 7. Hemocyanins in Combination with Adjuvants

The use of adjuvants with hemocyanin–hapten conjugates in experimental animals is a common practice to achieve higher antibody titers against the hapten. The immunogenicity of hemocyanins, when conjugated to small peptides or haptens, can vary depending on the chemical agent used for coupling. To ensure a strong and consistent immune response, these conjugates are often administered alongside Freund’s adjuvant. Freund’s Complete Adjuvant (CFA) contains mycobacterial components that activate pattern-recognition receptors such as TLR2 and TLR4, leading to local inflammation and enhanced antigen uptake by dendritic cells. In contrast, the Incomplete form (IFA) maintains a depot effect, allowing for sustained antigen release and prolonged stimulation of T and B cells. This combination induces higher antibody titers and more robust cellular responses compared to KLH alone, as demonstrated in animal immunization studies [137]. In humans, similar oil-in-water emulsions, such as Montanide ISA-51, which are considered safer alternatives to Freund’s adjuvant, have been shown to restore or enhance responses to subunit KLH [138]. Overall, the combination of KLH and Freund’s adjuvant exemplifies a synergism between a complex protein carrier and a potent adjuvant, enhancing both innate activation and adaptive immune priming.

The effects of CCH and FLH as carriers for P10, a mimetic peptide of GD2 (the main ganglioside component found in neuroectodermal tumors), were evaluated in combination with AddaVax. This MF59-like nanoemulsion is recognized as safe and effective in human vaccines. Both conjugates induced specific IgM and IgG antibodies against P10 without causing any toxic effects in mice. The results were comparable to those obtained with KLH-P10, which served as a positive control. Notably, FLH-P10 conjugates demonstrated superior antitumor activity compared to CCH-P10, as indicated by a significantly lower tumor growth rate and reduced tumor incidence in a preclinical model of melanoma [37]. In another study, the effects of these hemocyanins combined with Alum, AddaVax, and QS-21 were evaluated. The combination of QS-21 with hemocyanins produced the best results, as evidenced by a robust, specific humoral response predominantly characterized by an IgG2a antibody response. This combination enhanced the antitumor activity of hemocyanins in an orthotopic mouse model of oral squamous cell carcinoma, a highly aggressive and deadly cancer, reducing tumor development and improving overall survival [39].

## 8. Conclusions

Based on the evidence presented here, the adjuvant effects of hemocyanins can be partially attributed to the carbohydrates present in their structure. These carbohydrates activate APCs when hemocyanin binds to C-type lectins and Toll-like receptors. This interaction influences the intracellular destination of the APCs and regulates T-cell polarization. Specifically, hemocyanins can interact with MR and DC-SIGN, inducing signaling processes in APCs and triggering specific cytokine responses through the TLR-4 pathway. This generates an adjuvant effect that depends on recruiting MyD88 and TRIF (Figure 4). Additionally, hemocyanins exhibit remarkable conformational stability, resulting in a slow degradation process within cells and enabling long-term storage. They degrade through vacuolar and cytosolic pathways at a notably slow rate. This gradual degradation is associated with a delayed response in proinflammatory cytokine secretion, as well as with the expression of costimulatory molecules, which can lead to beneficial biomedical effects (Figure 5). This is especially important for vaccines, because using an antigen with delayed kinetics can enhance the magnitude, quality, and persistence of antibody responses [139].

When hemocyanins are used as carriers or adjuvants for peptide and hapten molecules, their stability allows the resulting conjugates to persist for an extended period within APCs, such as macrophages, dendritic cells, and B lymphocytes. This enables effective processing in the acidic environment of lysosomes and immunoproteasome with the subsequent presentation through MHC-II and MHC-I to CD4+ and CD8+ T lymphocytes, along with the secretion of specific antibodies by B lymphocytes. During this process, the Th-1 hemocyanin-dependent proinflammatory environment regulates and promotes a robust adaptive immune response, targeting both the ligands and hemocyanins. This leads to a beneficial bystander effect that helps overcome the immune tolerance typically established by tumors and other pathologies, accounting for the non-specific adjuvant effect of hemocyanins (Figure 5). This immune response is not harmful since mammals do not possess molecules similar to hemocyanins. As a result, immunotherapy using hemocyanins as carriers or adjuvants for peptides and haptens, or as adjuvants alone, faces no significant limitations or challenges.

The above mechanism also explains why hemocyanins can be administered exclusively as an adjuvant of antigen molecules, as is the case of OVA co-administered with FLH, where we have shown that hemocyanin not only induced classical MHC-II presentation to CD4+ T cells from OT-II mice, but also promoted cross-presentation by TLR4 signaling through the TRIF and MyD88 pathways, enhancing CD8^+^ T-cell priming from OT-I mice [125].

## 9. Projections

Hemocyanins have significant potential as immunomodulators due to their ability to interact with multiple innate immune receptors, particularly C-type lectin receptors like the mannose receptor MR and DC-SIGN, and Toll-like receptors, such as TLR4. These interactions promote a Th1-skewed proinflammatory response—a hallmark of trained immunity—in which monocytes, macrophages, and other innate immune cells undergo epigenetic and metabolic reprogramming after exposure to specific stimuli [140]. This overlap suggests hemocyanins may function as inducers of trained immunity, providing broad-spectrum innate protection and enhanced adaptive immune priming. These unique properties make hemocyanins promising candidates for vaccine adjuvants and immunotherapies that require sustained innate activation. Moreover, the conformational stability of hemocyanins enables their slow intracellular degradation, directing them to lysosomal and proteasomal degradation pathways. This process ultimately leads to the presentation of peptides in MHC-I and MHC-II. Importantly, most studies have focused on how hemocyanins affect the functionalities of innate immune cells (Figure 5). Still, we know very little about their impact on other critical aspects of immunity, such as the development of immunological memory, which is essential for effective vaccination, or their influence on effector cells like neutrophils, which play key roles in controlling infectious diseases.

Despite advances, the complex structure of hemocyanins has thus far prevented their successful expression in heterologous systems, including fusion proteins. As a result, their use remains limited to natural sources, which present challenges such as environmental impact, batch-to-batch variability, difficulties in achieving GMP-compliant production, and regulatory compliance. These limitations underscore the importance of identifying specific antigenic peptides within hemocyanins that retain adjuvant properties, offering advantages such as homogeneity, high reproducibility, and greater purity. One way we are exploring to assess these limitations involves identifying immunodominant antigenic peptides within hemocyanins that retain adjuvant properties for both MHC-I and MHC-II. These peptides offer advantages such as homogeneity, high reproducibility, greater purity, and the capability of being scaled up for GMP manufacture. This process will involve computational modeling, structural biology, immunology, and machine learning. Taken together, these properties and perspectives converge in the emerging concept of next-generation adjuvants—molecules inspired by natural hemocyanins but refined through protein engineering to enhance efficacy while fine-tuning their immunological profiles for specific contexts. In parallel, the establishment of a Hemocyanin Immune Dictionary integrating in vivo stimulation models, single-cell multi-omics, immunopeptidomics, and computational modeling could serve as a predictive framework to map hemocyanin-induced immune responses across cell types, tissues, and administration routes, ultimately guiding rational protein engineering for advanced hemocyanin-based adjuvants and immunotherapies with tunable potency, safety, and specificity. Altogether, these strategies bridge fundamental immunology, bioengineering, and translational research, charting a coherent path from natural biomolecules to their engineered successors.

Furthermore, given the exceptional conformational stability and slow intracellular degradation, hemocyanins also provide a unique opportunity to study antigen fate at single-cell resolution. Technologies such as the multimodal method DOGMA-seq, which simultaneously profiles DNA, RNA, and surface proteins, could be leveraged to trace hemocyanin uptake and persistence within draining lymph nodes after immunization. By tracking labeled hemocyanins from 1 to 3 days post-injection, it would be possible to identify which immune subsets internalize, process, or retain these molecules, and to correlate their localization with transcriptional, signaling, and metabolic states. When paired with diverse antigens or co-adjuvants, this approach could illuminate how adjuvant structure modulates antigen presentation and immune polarization, ultimately linking molecular persistence to protective immunity.

## Figures and Tables

**Figure 1 vaccines-13-01086-f001:**
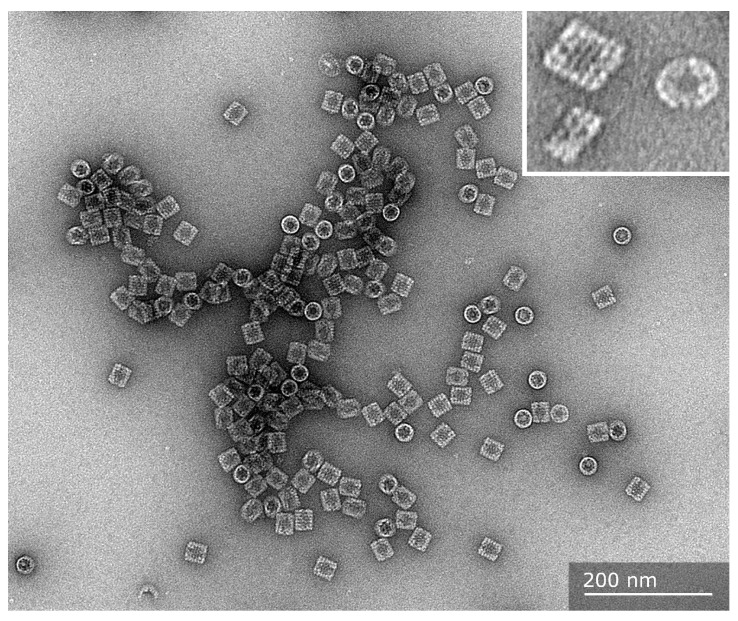
Transmission electron microscopy images of purified mollusk hemocyanin molecules. The purified molecules from *Concholepas concholepas* are shown from a top-view orientation (circles) and lateral views (rectangles). They display their basic quaternary structure, a cylinder didecamer, where solitary decamers are practically absent. In the inset, a higher magnification of a top view of a molecule reveals the internal collar region. The side views show a didecameric molecule formed by two decamers together, and the didecamer and decamer exhibit a three-tiered wall with FUs clearly visible. The sample was negatively stained with 2% uranyl acetate, and images were captured with a Talos F200C G2 electron microscope at the Advanced Microscopy Unit, Pontificia Universidad Católica de Chile.

**Figure 2 vaccines-13-01086-f002:**
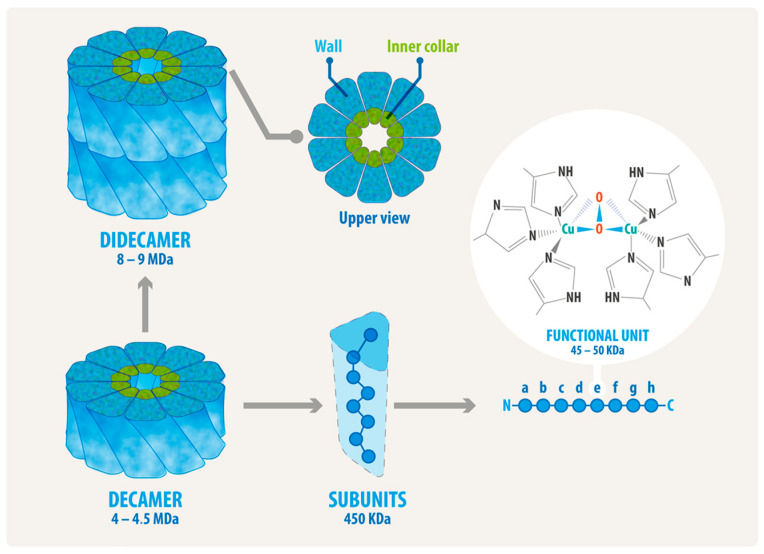
The structural levels of mollusk hemocyanins. The scheme shows that a decamer contains ten subunits and is the basic quaternary structure for gastropod hemocyanins. The subunit contains eight paralogous oxygen-binding globular folded domains called functional units, termed FU-a to FU-h, each of which contains a binuclear copper-binding site that binds molecular oxygen. In gastropods, as the mollusk species mentioned in this review, the decamers are assembled into didecamers. A top view of a molecule reveals the collar region.

**Figure 3 vaccines-13-01086-f003:**
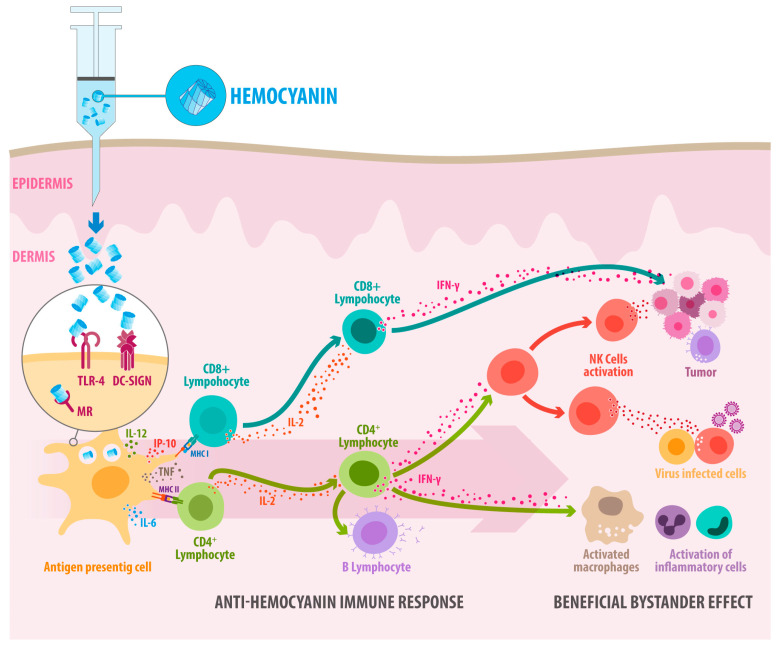
Hemocyanins are glycoproteins that act as non-specific adjuvants, stimulating the Th1-type immune response and thus generating a beneficial bystander effect. Experimental data support this scheme, summarizing the immune response of mammals against mollusk hemocyanins and their general immunomodulatory effects. When hemocyanin molecules are inoculated intradermically in mammals, they are endocytosed by the local antigen-presenting cells, both by micropinocytosis and receptor-mediated endocytosis, through innate immune receptors (MR, DC-SIGN, and TLR4) on the cell surface of dendritic cells, determining their processing fate to MHC-I and MHC-II processing pathways to activate CD8+ and CD4+ lymphocytes. Thus, hemocyanins drive gene expression toward the Th-1 phenotype, secreting IL-6, IL-12, IP-10, and TNF, which induces CD4+ T lymphocyte activation and INF-γ and IL-2 expression, concomitantly with B lymphocyte activation. Therefore, it generates a Th1 inflammatory milieu, producing a beneficial bystander effect. These cytokines activate macrophages and the attack of NK cells against tumors or virus-infected cells.

**Figure 4 vaccines-13-01086-f004:**
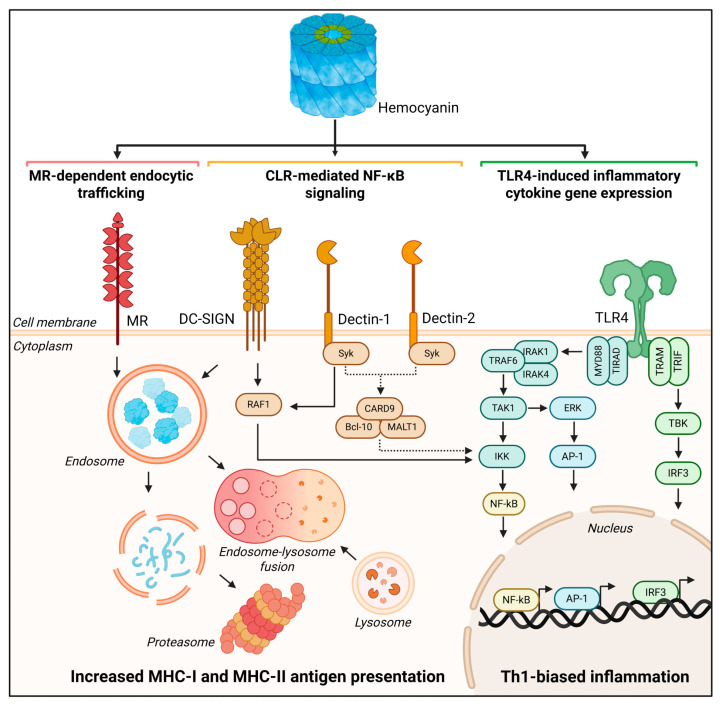
Innate immune receptors and signaling pathways activated by hemocyanins promote antigen presentation and Th1-skewed inflammation. Antigen-presenting cells internalize hemocyanins through endocytosis mediated by the mannose receptor (MR; red) and DC-SIGN (yellow), leading to trafficking through the MHC-I and MHC-II antigen processing pathways (solid arrows). DC-SIGN also activates the RAF1 signaling axis to promote NF-κB activation. Although direct binding between hemocyanins and Dectin-1 or Dectin-2 has not been demonstrated, downstream activation of Syk kinase and ERK1/2 suggests involvement of C-type lectin receptors (CLRs; yellow) in NF-κB activation (dotted arrows). TLR4 engagement by hemocyanins activates both MyD88- and TRIF-dependent pathways (green), leading to the nuclear translocation of NF-κB, AP-1, and IRF3, and promoting the expression of proinflammatory cytokine genes. These coordinated innate signaling events enhance antigen presentation and drive a Th1-biased immune response. This figure was designed using BioRender.

**Figure 5 vaccines-13-01086-f005:**
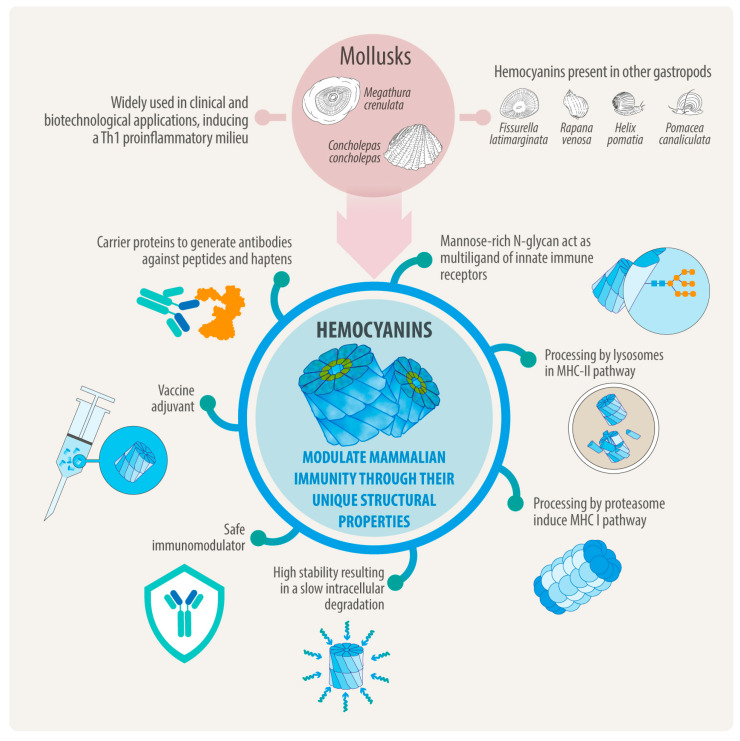
Biomedical and biotechnological applications of hemocyanins and the mechanism of action on the mammalian immune system. The diagram summarizes the main applications of hemocyanins and some structural properties that support their mechanisms of action.

## Data Availability

No new data were created or analyzed in this study.

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
