# Peer review of "Hemocyanins: Microscopic Giants with Unique Structural Features for Applications in Biomedicine"

_vaccines, 2025, doi:10.3390/vaccines13111086_

Round 1

Reviewer 1 Report

Comments and Suggestions for Authors

This is comprehensive yet concise review about hemocyanins, protein-based adjuvants including KLH. The manuscript introduces the family of mollusk and gastropod hemocyanins, their amazing quaternary structure and huge molecular weight, as well as their abundant glycanation. Next, the authors nicely explain how these specific features underly the functional effects of hemocyanins in APC (slow degradation, leakage into the cytoplasm, leading to cross-presentation; binding to and activation of CLRs and of the TLR4 pathway that forms the basis of the adjuvant activity). 

The review manuscript is very well written and interesting to read. The cited references contain quite a number from the same group, which is explained by their focus on this topic over the years. Still, the current literature on hemocyanins and the immunological pathways of adjuvant mechanisms is very well and balanced cited.

I have no significant criticism and recommend publication.  

Author Response

This is comprehensive yet concise review about hemocyanins, protein-based adjuvants including KLH. The manuscript introduces the family of mollusk and gastropod hemocyanins, their amazing quaternary structure and huge molecular weight, as well as their abundant glycanation. Next, the authors nicely explain how these specific features underly the functional effects of hemocyanins in APC (slow degradation, leakage into the cytoplasm, leading to cross-presentation; binding to and activation of CLRs and of the TLR4 pathway that forms the basis of the adjuvant activity). 

The review manuscript is very well written and interesting to read. The cited references contain quite a number from the same group, which is explained by their focus on this topic over the years. Still, the current literature on hemocyanins and the immunological pathways of adjuvant mechanisms is very well and balanced cited. 

I have no significant criticism and recommend publication.  

Our response: We appreciate the reviewer for carefully reviewing our manuscript and providing constructive feedback. We are grateful for their positive remarks and acknowledgment of our contributions to understanding the structure and mechanism of action of mollusk hemocyanins in mammals. We hope that you agree with the new version of the manuscript.

Reviewer 2 Report

Comments and Suggestions for Authors

The review is devoted to hemocyanin representatives, their structural properties, reception and biological effects. The description is clear, interesting and based on rich experience of authors in this field and presentation of numerous own experiments, results and references.  However, hemocyanin applications in biomedicine as mentioned in the title is poor reviewed and examplified.

A complete description of the structure, properties, activity, and effects of hemocyanins would only be clearly recognizable as their features when compared to alternative proteins, carriers, and adjuvants.

It is recommended that the review include a table listing known hemocyanins, their source, structure, molecular size, properties, spheres and examples of applications along with a list of references to key works. This would be a very useful and demonstrative addition to the text description.

Chapter 5 describes the higher stability of hemocyanin when processed in APC. However, the comparison is only with ovalbumin, which is practically unsuitable as a carrier protein for immunization. What other comparisons with widely used carrier proteins (BSA, tetanus toxoid, CRM197, etc.) can be made to demonstrate the unique properties of hemocyanins?

If hemocyanins have a pronounced adjuvant effect, as described, why do most researchers perform immunization with hapten conjugates with KLH using an adjuvant (Freund's)?

Author Response

The review is devoted to hemocyanin representatives, their structural properties, reception and biological effects. The description is clear, interesting and based on rich experience of authors in this field and presentation of numerous own experiments, results and references.  However, hemocyanin applications in biomedicine as mentioned in the title is poor reviewed and examplified.

A complete description of the structure, properties, activity, and effects of hemocyanins would only be clearly recognizable as their features when compared to alternative proteins, carriers, and adjuvants.

It is recommended that the review include a table listing known hemocyanins, their source, structure, molecular size, properties, spheres and examples of applications along with a list of references to key works. This would be a very useful and demonstrative addition to the text description.

 Our response: We thank the reviewer for their careful reading of our manuscript and constructive criticism, which has allowed us to improve it significantly. Based on these comments and suggestions, we have carefully modified the original manuscript and hope you agree with the new version. Below, we provide our point-by-point replies and explanations to the Reviewer. We have written our answers in italics to make them easier to read.

     1. The description is clear, interesting and based on rich experience of authors in this field and presentation of numerous own experiments, results and references.  However, hemocyanin applications in biomedicine as mentioned in the title is poor reviewed and examplified.

 Our response: We agree. We introduced a paragraph of this issue in Section 1 of the new version of the manuscript (lines 82 to 100).

 “KLH has garnered notable attention for its use in antitumor vaccines, which offer an alternative to conventional antiproliferative therapies, such as chemotherapy and radiotherapy, either alone or in combination with these treatments. One specific application has involved the use of KLH in autologous anti-idiotypic vaccines for B-cell lymphomas. In this approach, the variable regions of the immunoglobulin expressed on the tumorigenic B cell clone (Id) are chemically linked to KLH. Phase I/II clinical trials demonstrated that the Id-KLH vaccine combined with GM-CSF was effective in eliciting tumor-specific immune responses and achieving molecular remission in patients with stage 1 lymphoma (Bendandi et al., 2009; Rollig et al., 2011; Schuster et al., 2011). More recently, an immunotherapy approach combining an antimyeloma Id-KLH vaccine with vaccine-primed costimulated T cells has been reported, yielding a robust immune response with Id-specific humoral and cellular responses compared to the control with KLH alone (Qazilbash et al., 2022).  Furthermore, KLH also serves as an adjuvant that helps disrupt self-tolerance to tumor antigens. This immunotherapy involves generating ex vivo autologous tumor cell lysate -loaded dendritic cells and KLH that stimulate T-cell responses in cancer patients, showing promising outcomes in extending overall survival (Tittarelli et al., 2012;  2024). Beyond its anticancer applications, antimicrobial activity against the Gram-positive Streptococcus epidermis and the Gram-negative Escherichia coli has also been described in FUs of hemocyanins from Helix and Rapana snails (Dolashka et al., 2015).

  1. It is recommended that the review include a table listing known hemocyanins, their source, structure, molecular size, properties, spheres and examples of applications along with a list of references to key works. This would be a very useful and demonstrative addition to the text description.

Our response: Thank you for your suggestion. We agree. As indicated by the reviewer, we have incorporated Supplementary Table 1 in the revised version of the manuscript. Please refer to lines 141 to 143 for the mention of the table.

  1. Chapter 5 describes the higher stability of hemocyanin when processed in APC. However, the comparison is only with ovalbumin, which is practically unsuitable as a carrier protein for immunization. What other comparisons with widely used carrier proteins (BSA, tetanus toxoid, CRM197, etc.) can be made to demonstrate the unique properties of hemocyanins?

 Our response: We acknowledge this criticism; the investigation used OVA for several reasons. OVA has been used for comparison with hemocyanin to highlight the differences between them. OVA has a long history in research; its structure, composition, and immunogenic properties are well understood, and it features well-characterized immunodominant epitopes. For example, in veterinary medicine, immunization with capsular polysaccharide (CPS from Streptococcus suis serotype 2) conjugated to chicken OVA successfully induced protective immunity in pigs, with IgG antibody levels surpassing those from CPS–CRM197 conjugates (Kralova et al., 2022). Similarly, peptide-based vaccines are a new approach for Alzheimer’s disease, marked by amyloid-beta (Aβ) deposits, neurofibrillary tangles, and cognitive decline. Aβ1-10 peptides, especially the Aβ1-10-S8R conjugated to OVA or KLH, reduced Aβ buildup, lowered APP and BACE-1 expression, decreased pro-inflammatory cytokines by astrocytes and microglia, and increased synaptic proteins. These effects included higher IgG responses in the spleen, showing humoral immune activation (Park et al., 2025). These characteristics make OVA a reliable model for studying antigen processing and presentation. Antibody and T cell epitopes have been mapped for many strains of mice. As a result, OVA was one of the first antigen genes utilized as a transgene to create OT-1 and OT-II OVA-specific TCR transgenic mice.

In addition, OVA and hemocyanin can be obtained commercially with a high degree of purity. In contrast, tetanus toxoid is chemically inactivated, and the commercial product contains between 20% and 70% toxoid, along with numerous other protein contaminants from C. tetani, as identified by mass spectrometry (Möller et al., 2019). This contamination with additional proteins results in a highly immunogenic preparation. Still, its mechanism of action cannot be attributed to a single component, unlike OVA and hemocyanin. Thus, our studies indicate that not all foreign proteins exhibit the same intracellular processing kinetics and pro-inflammatory patterns as OVA, and that the effects are dependent on hemocyanin.

Finally, the variant of the diphtheria toxin protein, CRM197, would be particularly interesting to compare with CCH and KLH carrier/adjuvant, because CRM197 is not a glycosylated protein like hemocyanins, so its mechanisms of action may differ.

  1. If hemocyanins have a pronounced adjuvant effect, as described, why do most researchers perform immunization with hapten conjugates with KLH using an adjuvant (Freund's)?

Our response: Thank you for your pertinent comment. We have added a paragraph on this subject (lines 520 to 551)..

 Hemocyanins in combination with adjuvants

The use of adjuvants with hemocyanin-hapten conjugates in experimental animals is a common practice to achieve higher antibody titers against the hapten. The immunogenicity of hemocyanins, when conjugated to small peptides or haptens, can vary depending on the chemical agent used for coupling. To ensure a strong and consistent immune response, these conjugates are often administered alongside Freund’s adjuvant. Freund’s Complete Adjuvant (CFA) contains mycobacterial components that activate pattern-recognition receptors such as TLR2 and TLR4, leading to local inflammation and enhanced antigen uptake by dendritic cells. In contrast, the Incomplete form (IFA) maintains a depot effect, allowing for sustained antigen release and prolonged stimulation of T and B cells. This combination induces higher antibody titers and more robust cellular responses compared to KLH alone, as demonstrated in animal immunization studies (Le Moigne, 2011). In humans, similar oil-in-water emulsions, such as Montanide ISA-51, which are considered safer alternatives to Freund’s adjuvant, have been shown to restore or enhance responses to subunit KLH (Miller et al., 2005). Overall, the combination of KLH and Freund’s adjuvant exemplifies a synergism between a complex protein carrier and a potent adjuvant, enhancing both innate activation and adaptive immune priming.

The effects of CCH and FLH as carriers for P10, a mimetic peptide of GD2 (the main ganglioside component found in neuroectodermal tumors), were evaluated in combination with AddaVax. This MF59-like nanoemulsion is recognized as safe and effective in human vaccines. Both conjugates induced specific IgM and IgG antibodies against P10 without causing any toxic effects in mice. The results were comparable to those obtained with KLH-P10, which served as a positive control. Notably, FLH-P10 conjugates demonstrated superior antitumor activity compared to CCH-P10, as indicated by a significantly lower tumor growth rate and reduced tumor incidence in a preclinical model of melanoma [35]. In another study, the effects of these hemocyanins combined with Alum, AddaVax, and QS-21 were evaluated. The combination of QS-21 with hemocyanins produced the best results, as evidenced by a robust, specific humoral response predominantly characterized by an IgG2a antibody response. This combination enhanced the antitumor activity of hemocyanins in an orthotopic mouse model of oral squamous cell carcinoma, a highly aggressive and deadly cancer, reducing tumor development and improving overall survival [37].

Reviewer 3 Report

Comments and Suggestions for Authors

1、The abstract is not well - organized, and the description of the article's viewpoints is not clear enough. You can focus on discussing the features of this article.

2、In line 69, the text states that "recombinant expression of KLH has not been possible.". Then, can we discuss whether this will affect its application value?

3、In Figure 2, the picture of the binuclear copper binding site of the functional units can be enlarged, make the whole picture more vivid and rich.

4、In the fifth part, there is no indication of the reason for discussing this part. A few introductory sentences can be added.

5、At the beginning of part 6, the significance and purpose of researching the inflammatory pathway can be added.

6、In the discussion section, the author can elaborate on the future application prospects of hemocyanin and add some thoughts on how to apply hemocyanin.

7、The format of references needs to be unified. For example, in line 732, the format of "Br J Urol" is inconsistent with that of other journals.

Author Response

Thank you for your insightful observations and valuable comments, which we have thoroughly considered to improve our manuscript. We agree with the information you asked for, and additional explanations have been included. Also, as you suggested, we have revised the English with the help of the Author Service of Vaccines.  We hope that you agree with the new version of the manuscript. Below, you will find our responses to each point. For your convenience, our replies are highlighted in italics.

1、The abstract is not well - organized, and the description of the article's viewpoints is not clear enough. You can focus on discussing the features of this article.

Our response: We thank the reviewer for this constructive comment. In response, we have thoroughly revised the abstract to improve its organization, clarity, and logical flow. The new version now clearly presents the main viewpoints of the review by:

(i)    Introducing the current gap in adjuvant research and the need for better-defined immunomodulatory molecules.

(ii)   Highlighting hemocyanins as a distinctive class of natural protein-based adjuvants (PBAs) with well-characterized structural and immunological features.

(iii)  Summarizing their mechanisms of immune activation through innate receptor engagement and antigen presentation pathways.

(iv)  Discussing their biomedical relevance in infectious disease and cancer vaccine development.

(v)   Outlining key challenges (e.g., recombinant production, scalability) and future perspectives, including computational and structural approaches for next-generation adjuvant design.

These revisions make the abstract more coherent and explicitly state the scope, features, and objectives of this review.

2、In line 69, the text states that "recombinant expression of KLH has not been possible.". Then, can we discuss whether this will affect its application value?

Our response: We agree and appreciate your question because it allows us to clarify an important aspect of the use of hemocyanins. We have clarified this matter in the revised version of the manuscript (lines 106 to 114).

“However, this has not affected their primary applications. In fact, hemocyanins have advantages over other natural or therapeutically used antigens commonly used in experimental human immune challenges, such as varicella zoster or BCG. These traditional antigens do not allow for control over the degree, duration, and timing of primary exposure. The long history of using hemocyanins has established an excellent safety record, making them a safe and reliable model for immunomodulatory proteins. However, the limitations in the use of KLH are largely due to the methodological inconsistencies in their application, leading to a lack of standardized approaches in the design and conduct of early-phase drug development (Drennan et al., 2022).”

3、In Figure 2, the picture of the binuclear copper binding site of the functional units can be enlarged, make the whole picture more vivid and rich.

Our response: We agree. The correction in the figure 2 was actually done.

4、In the fifth part, there is no indication of the reason for discussing this part. A few introductory sentences can be added.

Our response: We agree. The requested introductory sentence has been added (lines 378 to 385), remaining as follows.

Research on the antigen processing of small proteins, such as ovalbumin (OVA), has been extensively conducted. However, similar studies on larger and structurally stable oligomeric metalloglycoproteins, such as hemocyanins, are still limited (Kotsias et al., 2019). As mentioned earlier, the glycosylation of hemocyanins contributes to their immunological effects by interacting with receptors involved in innate immunity. Interestingly, even deglycosylated hemocyanins and isolated subunits exhibit partial immunostimulant effects, indicating that additional structural features, such as quaternary structure or sequence, may contribute to their beneficial effects.”

5、At the beginning of part 6, the significance and purpose of researching the inflammatory pathway can be added.

Our response: We agree. The requested explanation was added (lines 464 to 471).  

“Investigating inflammatory pathways is crucial for comprehending the full spectrum of hemocyanin's immunomodulatory effects in mammals, including its potential as an adjuvant and non-specific anticancer agent. By exploring these mechanisms, we can gain insight into how hemocyanins influence immune responses, activating both innate and adaptive immunity, by interacting with innate receptors on antigen-presenting cells, inducing expression of cosignalling molecules,  triggering cytokine production,  and influencing T helper cell activation, expansion, and polarisation.” It is known that CLRs and TLRs activate distinct inflammatory pathways…

6、In the discussion section, the author can elaborate on the future application prospects of hemocyanin and add some thoughts on how to apply hemocyanin.

Our response: We agree. The requested future potential applications of hemocyanins were added (lines 616 to 642).

 “One way we are exploring to assess these limitations involves identifying immunodominant antigenic peptides within hemocyanins that retain adjuvant properties, both for MHC-I and MHC-II, offering advantages such as homogeneity, high reproducibility, greater purity, and the capability of being scaled up for GMP manufactureThis process will involve computational modeling, structural biology, immunology, and machine learning (Turner et al., 2023). Taken together, these properties and perspectives converge in the emerging concept of next-generation adjuvants—molecules inspired by natural hemocyanins but refined through protein engineering to enhance efficacy while fine-tuning their immunological profiles for specific contexts. In parallel, the establishment of a Hemocyanin Immune Dictionary integrating in vivo stimulation models, single-cell multi-omics, immunopeptidomics, and computational modeling could serve as a predictive framework to map hemocyanin-induced immune responses across cell types, tissues, and administration routes, ultimately guiding rational protein engineering for advanced hemocyanin-based adjuvants and immunotherapies with tunable potency, safety, and specificity. Altogether, these strategies bridge fundamental immunology, bioengineering, and translational research, charting a coherent path from natural biomolecules to their engineered successors.

Furthermore, given the exceptional conformational stability and slow intracellular degradation, hemocyanins also provide a unique opportunity to study antigen fate at single-cell resolution. Technologies such as DOGMA-seq, which simultaneously profiles DNA, RNA, and surface proteins, could be leveraged to trace hemocyanin uptake and persistence within draining lymph nodes after immunization. By tracking labeled hemocyanins from 1 to 3 days post-injection, it would be possible to identify which immune subsets internalize, process, or retain these molecules, and to correlate their localization with transcriptional, signaling, and metabolic states. When paired with diverse antigens or co-adjuvants, this approach could illuminate how adjuvant structure modulates antigen presentation and immune polarization, ultimately linking molecular persistence to protective immunity.

  1. The format of references needs to be unified. For example, in line 732, the format of "Br J Urol" is inconsistent with that of other journals.

 Our response: We thank the Reviewer for bringing this issue to our attention

Round 2

Reviewer 2 Report

Comments and Suggestions for Authors

The manuscript was supplemented in accordance with the recommendations madе